# Implementation of Newborn Hearing Screening in Albania

**DOI:** 10.3390/ijns9020028

**Published:** 2023-05-10

**Authors:** Andrea M. L. Bussé, Birkena Qirjazi, Allison R. Mackey, Jan Kik, André Goedegebure, Hans L. J. Hoeve, Ervin Toçi, Enver Roshi, Gwen Carr, Martijn S. Toll, Huibert J. Simonsz

**Affiliations:** 1Department of Otorhinolaryngology, Erasmus University Medical Center, 3015 Rotterdam, The Netherlands; 2Department of Ophthalmology, Erasmus University Medical Center, 3015 Rotterdam, The Netherlands; 3Department of Ear, Nose and Throat Diseases-Ophthalmology, University of Medicine of Tirana, 1000 Tirana, Albania; 4CLINTEC, Karolinska Institutet, 171 77 Stockholm, Sweden; 5Department of Public Health, University of Medicine of Tirana, 1000 Tirana, Albania; 6Independent Consultant in Early Hearing Detection, Intervention and Family Centered Practice, Ribble Valley BB7 2RA, UK

**Keywords:** universal newborn hearing screening, hearing loss, implementation study, Albania, follow up

## Abstract

Newborn hearing screening (NHS) was implemented in Albania in four maternity hospitals in 2018 and 2019. Implementation outcome, screening outcome, and screening quality measures were evaluated. Infants were first screened by midwives and nurses before discharge from the maternity hospital and returned for follow-up screening. Acceptability, appropriateness, feasibility, adoption, fidelity, coverage, attendance, and stepwise and final-referral rates were assessed by onsite observations, interviews, questionnaires, and a screening database. A post hoc analysis was performed to identify reasons for loss to follow up (LTFU) in a multivariate logistic regression. In total, 22,818 infants were born, of which 96.6% were screened. For the second screening step, 33.6% of infants were LTFU, 40.4% for the third, and 35.8% for diagnostic assessment. Twenty-two (0.1%) were diagnosed with hearing loss of ≥40 dB, six unilateral. NHS was appropriate and feasible: most infants are born in maternity hospitals, hence nurses and midwives could perform screening, and screening rooms and logistic support were supplied. Adoption among screeners was good. Referral rates decreased steadily, reflecting increasing skill. Occasionally, screening was repeated during a screening step, contrary to the protocol. NHS in Albania was implemented successfully, though LTFU was high. It is important to have effective data tracking and supervision throughout the screening.

## 1. Introduction

Newborn hearing screening (NHS) programmes have been successfully implemented in developed countries across the world to detect infants with hearing loss and to provide them with subsequent intervention. These programmes seem to be lacking more often in countries with lower health expenditure and Human Development Index [1,2]. Without an effective NHS programme in place, children with hearing loss (HL) remain undetected until delays in speech and language development are noticed by observant caregivers [3]. Children with HL that remains unaddressed during early childhood experience delays in spoken language development which can affect academic performance and social development [4,5].

In most countries, an NHS is performed shortly after birth, with the first screening step usually being performed before discharge from the maternity hospital. The average length of stay in a maternity hospital in Europe is 3.1 days after giving birth (range: 1.5–4.9 days) [6]. Screening before discharge ensures a high coverage, but residual fluid and debris in the ear after birth may cause some infants with normal hearing to fail the first screen [7,8].

A high proportion of infants lost to follow up (LTFU) is a serious obstacle to successful implementation of NHS even in high-income countries such as the USA [9]. It has been reported to be even more challenging in developing countries [10]. Common reasons for LTFU between screening steps include travel times and costs, parental educational level, parental awareness, organisation of the NHS, religion, ethnicity, premature birth, low birth weight, APGAR score, and NICU admission [11,12,13,14,15,16]. It is generally felt that when parents, screeners and hospital staff are better informed about NHS and the consequences of a newborn HL, attitudes towards the NHS improve and more families follow up with screening [14].

Infants may be LTFU more easily when false-positive results are high in the first screening step [2]. This was found to be more common in newly implemented screening programmes. Referral rates between the first and second screening step decreased when screeners gained more experience [14,17,18]. Efforts made to lower referral rates included repeating the test multiple times [14], screening at a later age, and additional training for screeners [17].

Within the EUSCREEN study (www.euscreen.org (accessed on 1 January 2023)), the cost effectiveness of paediatric vision and hearing screening programmes was compared among countries in Europe. A cost-effectiveness model (miscan.euscreen.org (accessed on 1 January 2023)) was developed to compare the cost effectiveness of hearing screening programmes in different countries while taking local circumstances into account such as demography and geography. It can assist in the introduction and modification of or disinvestment from screening programmes in countries or regions. Alongside the development of the model, NHS was implemented in four maternity hospitals in Albania in 2018 and 2019 [19], both to test the predictions of the cost-effectiveness model and to improve the cost-effectiveness model with field data from the implementation study.

Two previous attempts have been made to implement NHS in Albania. In the first study by Hatzopoulos et al. (2007) [11], 40% of well babies (WB) and 53% of infants admitted to the neonatal intensive care unit (NICU) did not follow up to diagnostic assessment after a referral from screening. In total, 0.18% of the NICU infants were diagnosed with HL. In the second study, by Beqiri and Nika (2015) [20], only 1.6% of the infants did not follow up with screening or diagnostic assessment and 0.21% of all infants were diagnosed with HL. Both screening programmes were not continued after the end of the study.

In our study, the newly implemented NHS programme was monitored, data on implementation outcome, screening outcome, and screening quality measures were collected, evaluated, and compared with other countries to determine facilitators and barriers to the implementation of NHS in Albania. The experience of the implementation of NHS in Albania may provide useful learning for other countries in the preparation for and implementation of NHS.

## 2. Materials and Methods

### 2.1. Preparation and Screening Protocol

Albania has a total area of 28,748 km^2^ and a population of 2,845,955 inhabitants [21]. Almost 40% of the Albanian population lives in rural areas [22]. About 28,000 infants are born in Albania annually [23], of whom approximately 28 to 84 (1 to 3 out of every 1000 infants) may be born with a permanent sensorineural HL ≥ 40 dB [24,25,26].

NHS was implemented in Albania in four maternity hospitals. These were two maternity hospitals in urban areas: Mbretëresha Geraldine (MG) and Koço Gliozheni (KG) in Tirana and two in rural areas: the local maternity hospitals in Pogradec and in Kukës (Figure 1). Within the current observational study, all infants born in one of the four maternity hospitals during the time of implementation (2018–2019) were eligible for screening. The EUSCREEN cost-effectiveness model was used to simulate and compare various possible NHS programmes in Albania [27], though, due to the lack of detailed data on the local conditions, and due to delays in the development of the model, the screening protocol had to be determined primarily on the expert opinion of the audiologists and the ENT surgeons involved in the EUSCREEN study. The following protocols were used:In the two clinics in Tirana and in the clinic in Progadec, all infants born healthy (WB) were screened with a three-step OAE-OAE-aABR protocol (OAE: otoacoustic emissions, aABR: automated auditory brainstem response). All infants admitted to the NICU in one of the clinics in Tirana and in Progadec and all infants born in Kukës were screened using a two-step aABR-aABR protocol;Only the maternity hospitals in Tirana had proper NICUs, while the NICUs in Kukes and Pogradec only admitted low birth weight or premature neonates without major problems. The neonates with severe pathology from all of Albania were sent for treatment to Tirana.

WB were first screened before discharge from the maternity hospital, between 24 and 48 h after birth (Figure 2). When indicated, the second screening step was scheduled two weeks after the first screening step and the third screening step was scheduled two weeks after the second screening step. Infants who did not receive a pass result in both ears on completion of screening were referred to Tirana for diagnostic audiological assessment and subsequent intervention. This took place at the Tirana University Hospital Centre (TUHC) or the Child Centre for Rehabilitation (CCR).

The implementation of NHS in Albania was not considered to be a clinical trial by the Albanian ethics committee nor was it a new procedure that was being used. Therefore, the implementation of the NHS did not need approval from an ethics committee. The study was, however, approved by the University of Medicine in Tirana and the Albanian Ministry of Health and Social Protection and parents were required to sign an informed consent form before their infant was screened.

Implementation was organised, coordinated, and facilitated by the local study coordinator (one of the authors, BQ), an otolaryngologist at the Tirana University Medical Centre and an experienced audiologist. During the preparatory phase, the local study coordinator made arrangements with the maternity hospitals. She acquired the screening devices, arranged equipment to fit screening rooms, and employed screeners. She also explored possibilities to expand screening to nationwide reach. The screening protocol and the objectives of the NHS programme that was implemented in Albania were described in detail before implementation started [27].

The screening was performed in the maternity hospital by nurses who were selected by the head of the maternity hospital or the NICU department and who were trained within the study to perform screening. The first screening step took place before discharge from the maternity hospital since the majority of infants are born in a maternity hospital in Albania [28]. Parents were informed through leaflets, posters, informational videos on television screens in the maternity hospital, and interviews on national television and radio. Written informed consent was sought from parents before the screening. Families of infants who received a failed outcome were asked to return to the maternity hospital for follow-up screening.

Infants who were referred after completion of screening were referred to the Tirana University Hospital Centre or Child Centre for Rehabilitation in Tirana for diagnostic audiological assessment. Diagnostic ABR devices were installed in both locations [19].

### 2.2. Organisational Changes Made to Improve Follow up in 2019 Based on Experiences in 2018

The main problem experienced in the first year of screening was the high number of infants that did not attend follow-up screening [19]. Consequently, a high number of infants who failed the first screen did not complete the entire screening protocol. Based on observations made in the first year, some changes were made to increase follow-up in the second year. All infants with a failed outcome on the first screen received an appointment to return for follow-up screening two weeks later. Screeners emphasised the importance and possible consequences of this follow-up screening. Parents were reminded to return for follow-up screening by a telephone call a few days before the appointment.

Small changes were made in the teams that conducted the screening. In the MG maternity hospital, screeners had to deal with a high workload. Following the resignation of one of the original screeners, two new screeners were appointed to manage the high number of infants born each day. In Pogradec, only a small number of infants were screened using the aABR, so one screener undertook all aABR screening to provide consistent screening quality. After the first training course, two refresher courses were organised. During these refresher courses, peer support for screeners was established, the screeners could share experiences and ask each other for help with the difficulties they encountered. The screeners kept in contact with each other throughout the two years of implementation. Peer support was maintained through phone calls and text messages. It is clear that this is a key area to consider in maintaining screener performance.

### 2.3. Database with Screening Outcome

Screeners filled out infant data and the pseudonimysed results of screening of each infant on a paper form. Additionally, the screeners administered a socioeconomic questionnaire to the parents (described previously [19,29]). The forms were transferred to a secure digital database. From this database, the following screening quality measures were identified and recorded for eligible infants in the study for each maternity hospital and each screener: coverage, attendance, referral rates between screening steps, and referral rate to diagnostic assessment. Attendance rates were calculated for each screening step. Infants for whom no follow-up screening was registered were considered LTFU. Infants who were registered as LTFU were phoned by the local screening team to record reasons for LTFU. These phone calls took place in two instances. The first after one year of screening and the second at the end of implementation. All infants LTFU at that point, for whom no reason was provided, were phoned. The database was compared to the local register in each maternity hospital to assess and improve accuracy. The screening data that were uploaded in the database could be accessed by the local study coordinator (BQ) who also performed most of the diagnostic assessments. LTFU was analysed in more detail in a post hoc analysis of the screening data collected in the database.

### 2.4. Outcomes from Implementation of the Screening Programme

To evaluate the barriers and facilitators of NHS implementation in Albania, implementation outcomes were evaluated, based on the framework by Proctor et al. [30] and Peters et al. [31]. Within this implementation study, the acceptability, appropriateness, feasibility, adoption, fidelity, coverage, attendance, stepwise, and final referral rate of the NHS were assessed. These outcomes were evaluated using observations of screening during onsite visits, interviews with screeners and parents, screener questionnaires, a socioeconomic questionnaire for the parents, screening outcome data collected through the database, and follow-up phone calls with parents of infants LTFU [19].

Three authors (AB, AG, and HH) visited the maternity hospitals before, during, and at the end of implementation to observe screening and assess the progress of implementation. During each of these visits, screeners and parents were interviewed and the questionnaires were distributed. Screeners were asked about their experiences with the screening protocol, the devices, the database, and their interactions with the parents. Parents were asked how they experienced screening and how they were informed about the screening programme. All screeners received a questionnaire containing 155 multiple-choice questions, covering seven domains: general information, attitudes of the screener towards hearing screening, hearing loss, parents of infants screened and their subcultures, individual features of the screener, and additional questions. Answers were compiled and compared across screeners, maternities, and in relation to the time when the questionnaire was filled out. Screeners conducted a short questionnaire among parents that contained 13 questions on sociodemographic and socioeconomic subjects. Answers were collected in the study database and used in the post hoc analysis to identify reasons for LTFU.

### 2.5. Post Hoc Analysis of the Database to Identify Reasons for LTFU

To gain more insight into possible reasons for LTFU, a post hoc analysis was performed based on infant screening outcomes registered in the database. The outcome variable was the LTFU between screening steps 1 and 2 (yes/no).

It is important to recognise that infants and their families have individual characteristics that may predict their likelihood to be LTFU. Infants were grouped by the screener, and screener characteristics may also influence whether infants are LTFU. Two levels of analysis were thus considered: the individual level and the screener level. At the individual level, the predictor variables applied to either the mother/family or the infant. Some maternal/family variables were collected through the socioeconomic questionnaire, and these variables included the family’s economic status (5-point scale: very bad to very good), the mother’s overall health status (5-point scale: very bad to very good), and the mother’s age. Other family variables were the location of the family’s home, as reported by the parents (urban or rural), and the duration of time needed to travel from the family’s home to the screening facility (in hours), calculated through spatial analysis. Infant variables were registered in the database by the screeners and included the year of birth (2018 or 2019), the gender, the duration of pregnancy (in weeks), the number of risk factors for HL (zero or at least one) [32], and whether the infant failed the screening in one or both ears. At the screener level, the predictor variables were the protocol used by the screener, the total number of infants screened, and the overall referral rate for each screener.

#### 2.5.1. Geocoding Travel Time

To maintain anonymity, exact residential addresses were not obtained from the family apart from their hometown. A total of 392 hometowns were listed in the database, out of which 95 hometowns were listed among infants who were referred from screening step 1. The geographic information system (GIS) software QGIS v. 3.14 was used for geocoding and to perform the spatial analysis. All hometowns and four screening institutions were geocoded to spatial coordinates (x, y). The fastest travel time by car, in hours, was calculated from each family’s hometown to their respective screening institution using the Open Route Service plugin function for road-network analyses.

#### 2.5.2. Analysis

To investigate the variables that significantly predict which infants will be LTFU between screening steps 1 and 2, a hierarchical multivariate logistic regression analysis was performed in SPSS (v. 27). This type of analysis is most appropriate to perform when individuals are grouped by another factor, such as a screener. The final model contains the variables that best predict which infants are LTFU between screening steps 1 and 2 at a significance level of less than 0.05.

To select the variables for inclusion in the multivariate logistic regression analysis, univariate analyses using likelihood ratios were performed for each predictor variable. Predictor variables with a significance level of less than 0.1 in the univariate analysis were accepted into the multivariate model. Categorical variables with more than two categories were dummy coded. Categories were merged in two instances of quasi-complete separation of categorical variables due to the low number of infants in each group. The categories of very bad and bad were merged for the variables mother’s economic situation (n = 2 and n = 9) and health status (n = 3 and n = 125). Infants were excluded from the analysis if they passed screening and therefore were not required to attend rescreening or if they were referred from screening directly for diagnostic assessment. Missing covariate data were imputed using multiple imputations with chained equations.

## 3. Results

### 3.1. Outcome of the Screening in Infants

All 22,818 infants born were invited for screening and 22,051 infants had a screening result documented for the first screening step. Coverage in the first screening step in the maternity hospital was high (96.6% on average) throughout the two years of implementation. Out of the infants who participated in the first screen, 21,490 (97.5%) infants completed the entire screening protocol and received either a pass result in one of the steps or were referred to diagnostic assessment. LTFU was 519 out of 1546 (33.6%) between the first and second screening steps and 42 out of 104 (40.4%) between the second and third screening steps. These data are described in detail in Figure 3. Of the 81 infants who were referred for a diagnostic assessment, 52 (64,2%) attended. Twenty-two infants (0.1% of 22,051 infants screened at least once) were diagnosed with an HL of 40 dB or greater, of which 6 had a unilateral HL.

In the first screening step, the parents of 28 infants declined, 519 infants were LTFU after referral to the second screening step, and 42 after referral to the third screening step. Reasons for not attending a screening step given by the parents were registered in the screening database. These reasons include health issues (11); the infant died (24); the infant was discharged before screening took place (2); the infant was screened in another location (21); travel distance (88); economic reasons (5); parental refusal (147, of which 39 parents believed their infant could hear); parents could not be contacted (47); and for a number of infants the reason remained unknown (244). The ‘unknown’ category included parents who had not attended screening without providing a reason. Parents of infants that were LTFU were phoned at two different times during the study to obtain reasons for the LTFU. In addition to the reasons listed above, reasons provided during these telephone calls were: religious reasons; not understanding the purpose of the screening programme; no contact information; parents moved away; parents said they were not invited for screening or that no screening staff were present in the hospital when they brought their infant for screening.

Due to the low population density in the rural provinces, it was anticipated that in Kukës and Pogradec, long travel times between the home and the maternity hospital would negatively influence the parents’ willingness to return for follow-up screening. During the study, it became clear that even parents residing in Tirana had to travel for many hours to reach the maternity hospital, as mothers from all over Albania give birth in Tirana, which made it more difficult for them to travel back for screening once they had returned home.

The results of the screenings are reported based on data obtained through the study database. Data are reported for all infants, both WB and infants admitted to the NICU, combined due to inadequate data recording, as not all screeners recorded NICU/WB in the database. For some infants, a NICU stay was recorded in the database while an OAE-OAE-aABR protocol was used and vice-versa. Furthermore, several issues occurred with the registration of screening outcomes in the database. Some were caused by technical problems such as duplicate files and others were caused by mistakes that were made when filling out the database forms. For example, for some infants, no follow-up screen was registered, while parents indicated having attended this screen. Some parents did not attend the scheduled appointment but returned for screening at another time, and a number of screeners indicated they did not complete the database in this case. All infants with a failed outcome in the first or second screening step for whom no follow-up screen was registered in the second or third screening step were considered LTFU. The database was checked for inconsistencies and obvious mistakes were corrected.

Some of the information on the results of diagnostic assessment and early intervention with hearing aids or through family support was missing. Since different institutions and professionals were involved in follow-up after the screening, it was difficult to gather data on all infants who were referred after the screening.

#### 3.1.1. LTFU per Screener and Maternity Hospital

Over the course of two years, 22 screeners were involved in the implementation of NHS (Table 1). Each screener screened an average of 1574 infants in the MG maternity hospital (range: 624–2620), an average of 1110 in the KG maternity hospital (range: 523–1467), an average of 206 in Pogradec (range: 182–234), and an average of 412 in Kukës (range: 298–595). The referral rates and LTFU varied across screeners, as displayed in Table 1. The proportion of infants that were LTFU per screener between the first and second screening step ranged from 0% to 93%. LTFU was highest in the MG maternity hospital, where most infants were born and the workload per screener was the highest. For two screeners, LTFU was as high as 78.7% and 93.0%. These two screeners accounted for 46.5% of all infants LTFU from the MG maternity hospital. This was partly due to inadequate data recording: one of these two screeners resigned within the first year and it is possible that some of the appointments for the second screening step she made were not registered or followed up by the other screeners.

Average referral rates across maternity hospitals were not correlated to LTFU. Although the number of infants born in the rural maternities in Pogradec (OAE-OAE-aABR protocol) and Kukës (aABR-aABR protocol) were similar, the average referral rate was much higher in Pogradec (24.4% of infants screened were referred to screening step 2 and 1.4% of all infants were referred to diagnostics) than in Kukës (7.3% of infants were refer red to screening step 2 and 0.6% of all infants were referred to diagnostics). LTFU rates for screening step 2 were low in both Pogradec (19%) and Kukës (18%).

#### 3.1.2. Post Hoc Analysis of the Database to Identify Reasons for LTFU

Out of the total of 22,051 infants screened and who had a result for the first screening step documented, 1559 failed the first screening step, 13 of which were referred directly to diagnostic assessment and 1546 were referred to the second screening step. Five hundred and nineteen infants (33%) did not follow up with the second screening step, and 1027 infants were screened of which 881 passed and 146 failed.

Results of the univariate analyses for LTFU between screening steps 1 and 2 showed that the following variables were significant predictors: region (urban/rural) and duration of travel from the family home to the hospital (Table 2). These predictors were incorporated into the hierarchical multivariate logistic regression model. Out of the 1546 infants eligible for inclusion in the multivariate analysis, predictor variable data were missing for 95 infants and these data were imputed. In the multivariate model, region (urban/rural) was not significant (*p* = 0.41). The analysis did not show a significant difference in LTFU between the first and second years of screening. The changes made after the first year of screening did not target specific screeners with high LTFU. The only significant predictor variable was the travel duration from the family home to the hospital (*p* < 0.001). Infants of parents who had to travel longer were more likely to be LTFU (odds ratio 1.61, 95% CI 1.39–1.86). For every additional minute of travel, the likelihood of LTFU for screening step 2 increased by 1%.

### 3.2. Outcomes of Implementation of the Screening Programme

Results for each implementation outcome measure are summarised in Table 3. Regarding acceptability, both parents and screeners considered NHS to be important and wanted to participate in the programme. Despite having experienced several difficulties while setting up the NHS, screeners reported that NHS was important and that it was suitable for Albania. NHS was considered appropriate since it could be implemented within the existing organisation of neonatal preventive healthcare in Albania. Audiological diagnostic assessment was made available for all infants who failed screening and early intervention for all infants who were diagnosed with HL. It was feasible to employ nurses and midwives to screen infants before discharge from the maternity hospital. However, it was more difficult to find a quiet room to perform screening in the MG hospital. Adoption was good and take up of the new role of screener was well accepted. All nurses who were invited participated in screener training and all performed screening. Observations of the screeners showed that the organisation of screening and individual screening skills improved throughout the two years of implementation. Regarding fidelity, most screeners adhered to the screening protocol; however, in the initial stages, some screeners in the MG maternity hospital repeated the screening test multiple times during one screening step in order to obtain a pass result. The protocol prescribed that the test could be repeated only once after an initial failed outcome. This issue was addressed during the yearly refresher training course and demonstrates the importance and the need for good monitoring using the screening database.

## 4. Discussion

The experience of implementing NHS in the Albanian context has highlighted a number of key areas of learning that could support other countries that are looking to implement NHS. The programme was successful in reaching a high proportion of infants. Screening performed by trained nurses and midwives before discharge from maternity hospitals, shortly after birth worked very well with high coverage in the first screening step (96.6%). Referral rates decreased steadily when screeners gained experience. However, LTFU between screening steps and to diagnostic assessment was the largest problem throughout the two years of implementation.

High coverage was achieved by performing the first screening step before discharge from the maternity hospital. The programme benefitted greatly from strong leadership from the local study coordinator (one of the authors, BQ), who understood the context in which the implementation took place. She played an integral role in training, supporting the screeners, and informing parents on the existence and importance of screening, which led to high participation. These efforts resulted in motivated screeners and well-informed parents who were willing to participate in the programme and this highlights the importance of sharing information with stakeholders.

LTFU was the main problem throughout the two years of screening. LTFU is not only a serious problem in low-income countries and middle-income countries but was also found in NHS programmes in high-income countries [33,34,35]. In our study, LTFU was highest in the MG maternity hospital where the most infants were born and the workload per screener was the highest. Eight nurses screened part time in the MG maternity hospital, in addition to their regular duties. Remarkable is the wide variation of LTFU rates among screeners (0–93%). The fact that one of the screeners resigned in the first year may be partly responsible for this, as some of the appointments for the second screening step may not have been followed up. The large variation of LTFU between screeners emphasizes the importance of good data tracking and supervision. In addition, the question could be asked whether LTFU would have been lower with fewer, but full-time and dedicated, screeners.

Between the first and the second screening step, 33.6% of infants were LTFU, 40.4% between the second and the third, and 35.8% on referral to diagnostic assessment. A high number of infants that were LTFU implies that some infants with HL remained undetected and untreated. LTFU was related to the travel time between the infants’ homes and the maternity hospital. Families who had to travel longer were more likely to be LTFU (odds ratio 1.61, 95% CI 1.39–1.86). The likelihood of an infant being LTFU increased by 1% for every additional minute they had to travel to the maternity hospital. Out of all infants who received a failed outcome in screening step 1 and who were invited to return for the second screening step, only 39% reached screen completion. This means that some infants with hearing loss were likely not diagnosed or did not receive timely intervention.

Travel time between hometown and maternity hospital was a significant predictor for infants LTFU between the first and second screening step even when the variation in screener performance was taken into account. When screening is continued and extended to nationwide reach, more maternity hospitals closer to the parents’ homes will be able to provide NHS. LTFU could be further reduced by combining screening with other health care appointments, planning screening steps closer together, reducing the number of screening steps, or performing two screening steps during the same appointment. However, these measures could reduce the specificity of the second test [14,17,18]. These solutions were also reflected in the predictions of the cost-effectiveness model that was developed within the EUSCREEN study [27].

A total of 22 of the 22,051 infants were diagnosed with HL of 40 dB or greater of which six had a unilateral HL. Although more infants with HL were expected to be found based on the literature [11,20,36,37] this may be explained by the high proportion of infants lost to follow up between screening steps and to diagnostic assessment as only 39% of infants with a failed outcome in screening step 1 reached screen completion. It is, however, possible that some infants completed more screening steps but were lost to documentation instead of LTFU. This could be resolved by using a database that can track infants through screening, diagnostic assessment, and intervention. An integrated data repository would also help identify false-negative subjects, enabling the hospital to identify the need for corrective measures (e.g., retraining personnel and updating devices).

As reported in our first paper about the implementation study (19), and as reported by screeners in interviews, referral rates after the first screening step were high initially but decreased as screeners gained more experience, indicating the dedication and motivation of the screeners. In the first year of screening, it took more time for the screeners in Pogradec and Kukës to decrease referral rates after the first screening step. This can be related to the low number of births in these maternity hospitals compared to Tirana. At the end of 2019, referral rates after the first screening step in Kukës were comparable to MG and KG in Tirana. However, referral rates in Pogradec remained higher throughout the two years of implementation. This could be related to the low number of births, training, or the fact that screeners in Pogradec had to learn both the OAE and aABR techniques. The decrease in the referral rate in the MG hospital in Tirana may partly be explained by some screeners repeating the screening test multiple times within one step. This issue was addressed and screeners could repeat the test a maximum of two times. Going forward, it will be incorporated into the training, as well as in the yearly refresher course.

When screening will be continued and extended nationwide, screening will be performed in all maternity hospitals in Albania to obtain high coverage and to provide a screening location close to the family’s home to reduce travel distance. It could be beneficial to implement a single-device protocol in maternity hospitals with a low number of births to improve screener experience and achieve low referral rates. This could be further improved by hiring a dedicated screener. The results of this study emphasise the importance of a reliable monitoring system to keep track of all eligible infants and their screening outcomes, to monitor the screening programme, to assure the quality of the screening, to document the results, and to adjust protocols based on context to improve the outcomes of the screening programme.

The implementation of screening as part of the EUSCREEN study ended on 31 December 2019. Efforts were made to improve screening outcomes throughout the two years of implementation. Progress was made, however some issues remained at the end of the study. It takes time for a screening programme to run effectively, especially when setting up a new programme [38]. Plans were made to gradually extend the NHS throughout Albania. However, due to the COVID-19 pandemic, money that was allocated for the extension of the NHS had to be used elsewhere, which slowed down the plan to extend the NHS nationwide. This plan was continued when priorities shifted from controlling the global pandemic to other healthcare interventions.

A weakness in our study was that the screening protocol for NHS in Albania was not really based on the predictions of the cost-effectiveness model of the EUSCREEN Study. We had anticipated having the first version of the cost-effectiveness model up and running after six months (mid-2017) which would then predict the optimal screening protocol for Albania three months later. That proved to be impossible, mainly due to the lack of available data. The lack of available data has, however, by itself become a main outcome of the EUSCREEN project. Comparison of the cost effectiveness of screening programmes across borders is hampered greatly by the lack of available data [39] and the reimbursement of screening programmes should be made conditional on providing (anonymized) screening data, to be able to compare their cost effectiveness. Albeit later than anticipated, the EUSCREEN cost-effectiveness model is up and running, however, and available at miscan.euscreen.org.

In a recent analysis with the EUSCREEN cost-effectiveness model [40] of the now available data of the implementation of NHS in Albania, with the high percentages of LTFU found, it was shown that a two-step NHS screening protocol would have been more cost effective than the three-step screening protocol in well babies under these circumstances. Interestingly, in our study, screeners referred some 30% of the ‘fails’ after step 2 directly to diagnostics, skipping step 3 (Figure 3), seemingly aware of accumulating LTFU with more screening steps.

## Figures and Tables

**Figure 1 IJNS-09-00028-f001:**
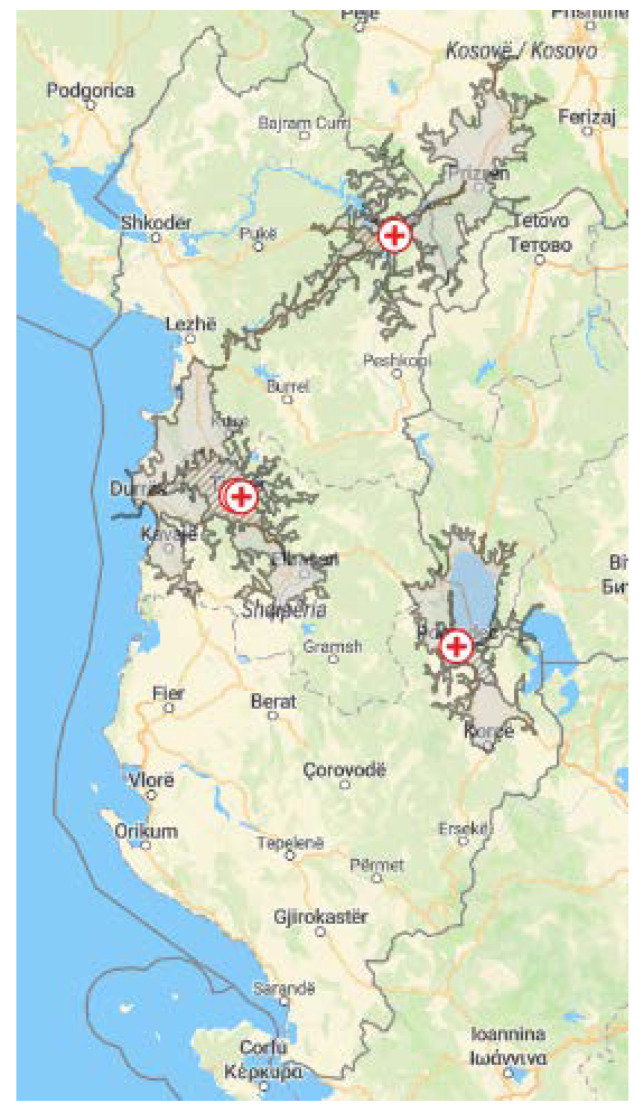
Map of Albania. ⊕ marks the four maternity hospitals: two in Tirana, one in Pogradec, and one in Kukës. The striped area marks the region around each maternity hospital within a travel time of 30 min by car. The grey area marks the region around each maternity hospital within a travel time of 60 min by car.

**Figure 2 IJNS-09-00028-f002:**
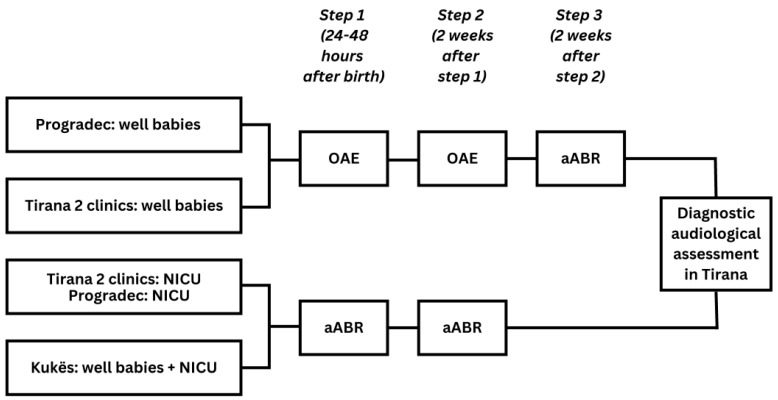
Screening pathway in Tirana, Pogradec, and Kukës.

**Figure 3 IJNS-09-00028-f003:**
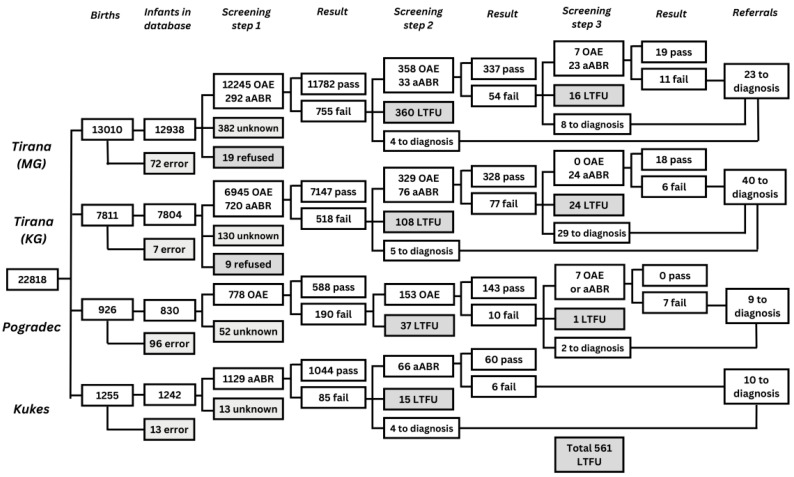
Flow chart depicting the number of infants screened for each screening step, the number of infants with a pass and fail result, the number of infants referred to diagnostic assessment, and the number of infants LTFU in all four maternity hospitals in Albania. ‘MG’: Mbretëresha Geraldine, maternity hospital in Tirana; ‘KG’: Koço Gliozheni, maternity hospital in Tirana; ‘OAE’: number of infants screened with OAE; ‘aABR’: number of infants screened with aABR; ‘error’: infants who were not recorded in the database; ‘unknown’: infants for whom no results were recorded in the database; ‘Pass’: infants who reached the threshold on the screening test; ‘Fail’: infants for whom the test failed or did not reach the threshold; ‘Refer to diagnostics’; refer for diagnostic assessment in Tirana; ‘LTFU’: infants who did not participate in subsequent screening due to various reasons. All infants admitted to the NICU in one of the two clinics in Tirana and in Progadec and all infants born in Kukës were screened using a two-step aABR-aABR protocol.

**Table 1 IJNS-09-00028-t001:** For each of the 22 screening nurses, the number of infants screened in step 1, referred to step 2, LTFU, referred to step 3, LTFU, referred to diagnostics and LTFU is listed below. For a few infants, no screener was registered, so they were not included in the table. ‘MG’: Mbretëresha Geraldine, maternity hospital in Tirana; ‘KG’: Koço Gliozheni, maternity hospital in Tirana; ‘P’: Progadec, maternity hospital in Progadec; ‘K’: Kukës, maternity hospital in Kukës. Note the large differences in LTFU, especially in the largest, busiest, maternity clinic in Tirana.

	Screen Step 1	Refer Step 2	LTFU Step 2	Refer Step 3	LTFU Step 3	Refer Diagn.	LTFU Diagn.
MG 1	2347	128	99 (77.3%)	2	0	7	5
MG 2	695	30	19 (63.3%)	3	2	1	0
MG 3	2682	183	35 (19.1%)	15	5	3	1
MG 4	905	19	8 (42.1%)	7	0	4	2
MG 5	1400	62	34 (50.9%)	4	0	1	0
MG 6	752	71	66 (93.0%)	2	0	1	0
MG 7	2407	202	64 (31.7%)	18	7	4	0
MG 8	1750	64	35 (54.7%)	3	2	6	1
KG 1	533	42	2 (4.8%)	7	2	6	2
KG 2	1470	187	41 (21.9%)	9	3	8	3
KG 3	1364	31	14 (45.2%)	6	0	6	0
KG4	1170	90	22 (24.4%)	13	3	5	0
KG5	1309	85	18 (21.2%)	21	14	5	4
KG6	1032	16	2 (12.5%)	9	2	3	2
KG7	926	72	9 (12.5%)	12	0	7	3
P 1	234	61	17 (27.9%)	5	1	3	2
P 2	195	41	7 (17.1%)	1	0	1	0
P 3	185	46	8 (17.4%)	1	0	3	2
P 4	216	42	5 (11,9%)	3	0	2	1
K 1	597	42	10 (23.8%)			2	1
K 2	347	19	0 (0%)			2	1
K 3	298	28	6 (21.4%)			2	1

**Table 2 IJNS-09-00028-t002:** Univariate and multivariate logistic regression analyses show the individual and screener-level predictors of loss to follow up between screening steps 1 and 2. Predictors significant at *p* < 0.1 in the univariate analysis were included in the multivariate analysis. Odds ratios are presented with 95% confidence intervals. The reference category or continuous variable unit is indicated in brackets.

Variable	Univariate Analysis		Multivariate Analysis	
	Odds Ratio (95% CI)	*p*-Value	Odds Ratio (95% CI)	*p*-Value
*Individual-level predictors*				
Family economic status (ref: Very good)	--			
Good	0.81 (0.45–1.45)	0.47	--	
Moderate	0.77 (0.44–1.33)	0.35	--	
Bad/Very bad	0.94 (0.45–1.45)	0.87	--	
Mother’s health status (ref: Very good)	--			
Good	1.0 (0.69–1.44)	1.0	--	
Moderate	1.35 (0.80–2.28)	0.27	--	
Bad/Very bad	0.97 (0.25–3.87)	0.97	--	
Mother’s age (Years)	1.00 (0.98–1.03)	0.54	--	
Region of family home (ref: Rural)	--			
Urban	0.78 (0.60–1.02)	0.07	0.89 (0.68–1.17)	0.41
Travel time (Family home to screening institution, hours)	1.62 (1.41–1.87)	<0.001	1.61 (1.39–1.86)	<0.001
Infant sex (ref: Girl)	--			
Boy	0.96 (0.75–1.22)	0.73	--	
Screening year (ref: 2019)	--			
2018	1.11 (0.84–1.49)	0.45	--	
Duration of pregnancy (Weeks)	0.99 (0.93–1.06)	0.80	--	
Risk factors (ref: Yes)	--			
No	1.15 (0.74–1.79)	0.54	--	
Screen 1 result (ref: Unilateral fail)	--			
Bilateral fail	1.05 (0.82–1.35)	0.69	--	
*Screener-level predictors*				
Test method (ref: Both OAE, aABR)	--			
aABR only	0.33 (0.06–1.74)	0.19	--	
Infants screened (Total number)	1.00 (1.00–1.00)	0.14	--	
Referral rate (%)	0.96 (0.89–1.03)	0.27	--	

**Table 3 IJNS-09-00028-t003:** Implementation measures and screening quality measures, definitions, operationalisation and outcomes [31]. Each outcome is followed by a two-letter code indicating the source: screener interviews (SI), screener questionnaire (SQ), parental interviews (PI), onsite observation of screening (OS), onsite observation of diagnostics (OD), onsite observation of intervention (OI), follow up phone calls with parents of infants lost to follow up (PP), database with screening outcome (DS), post hoc analysis of database (AD).

Measures and Definitions	Operationalisation	Outcomes and Sources
***Acceptability:*** the extent to which the programme is considered agreeable, palatable, or satisfactory by staff or other stakeholders.	*How important do screeners, parents, doctors, audiologists, healthcare administrators, and policymakers think NHS is for Albania?*	Screeners considered hearing screening important so that hearing loss in infants is detected early. (SI, SQ) Screeners indicated that all infants in Albania should have access to NHS. (SI, SQ) Parents thought hearing screening was important for their child after having received information on screening. (PI) Parents sometimes felt anxious about screening. (PI)
***Appropriateness:*** the perceived fit and relevance for stakeholders and the setting in which it is implemented.	*Is NHS relevant in current healthcare in Albania when compared to other healthcare priorities?*	NHS can be implemented within the existing organisation of neonatal preventive healthcare. (SI, SQ, OS) Most births in Albania take place in a maternity hospital, which facilitates the first screening performed before infants are discharged. (OS)
***Feasibility:*** the extent to which it can be successfully used or carried out within a given setting, its practicality.	*Can NHS be practised successfully in maternity hospitals by nurses and midwives?* *Is audiological diagnostic assessment and intervention available to all infants screened positively or referred?*	Performing NHS in the maternity hospital provides easy access to infants since, in Albania, the majority of infants are born in maternity hospitals. (SI, OS) Nurses and midwives who were already employed by the maternity hospital were able to perform screening in addition to other tasks. (SI, SQ, OS) Screening rooms were available at the maternity hospitals. (SI, OS) It was challenging to find a quiet room for screening in the maternity hospital with the largest number of births (MG). (SI, OS) When screening is continued, the budget should be made available to acquire additional devices. (OS) Fragility of the OAE probes and aABR electrode cables made them vulnerable to inexperienced handling. (SI, OS) Intervention with hearing aids was available to all infants with a confirmed permanent HL; furthermore, family intervention was made available by training a multidisciplinary team that included speech therapists, psychologists, and paediatricians. (OI) Parents indicated that they experienced difficulties returning to the maternity hospital because of long travel times, because they thought their infant could hear, or because the infant had other health issues. (PP)
***Adoption:*** the intention of the stakeholders to participate in the programme.	*How many parents of infants invited for screening agreed to participate in the programme? Did screeners agree to partake in NHS and integrate it into their daily routine? Do screeners want to put in more effort to detect infants with HL by extending their knowledge and skills? Did screeners change their attitude towards NHS? How important is it for the screeners to detect children with hearing loss that can then be treated?*	Maternity hospitals facilitated NHS by providing a screening room, logistic support, and time for the screeners to participate in the programme. (SI, OS) All screeners, who were trained during the project performed screening. (SI, SQ, OS) Screeners wanted to improve their screening skills. (SI, SQ) Throughout the two years of implementation, referral rates decreased steadily, reflecting increased screening skills. (DS, reported in [19]) Parents agreed with screening. (PI) However, they did not return for follow up screening. (DS) Database administration was integrated into the daily routine reasonably well, given that the programme had just started. (DS)
***Fidelity:*** the extent to which the agreements and prescribed protocols were adhered to during the implementation.	*Was screening carried out as prescribed in the screening protocol? How was the screening monitored and supervised? How accurately was administration performed?*	The screening protocol was followed. (SI, OS, DS) However, in the MG maternity hospital, screeners sometimes repeated OAE screening several times during the first screening step to obtain a pass result. (SI, OS) Some infants admitted to the NICU were not screened according to the NICU protocol. (SI, DS) Mistakes were made while filling out the study database forms, for example, the day of birth or the day of screening. (DS) Contact information was not registered for some infants, making it impossible for the screeners to contact these parents for follow up. (DS, PP) The MG maternity hospital did not have enough screeners or working hours for the number of births each day and supervision was insufficiently strict. (OS, DS, PP) Two screeners had 78.7% and 93.0% LTFU between the first and second screening steps. This accounted for 46.5% of infants LTFU to the second screening step in the MG maternity hospital. (DS)
***Coverage:*** the proportion of eligible infants that was screened.	*What proportion of eligible infants was screened?*	All infants born in 2018 and 2019 in one of the four maternity hospitals in Tirana, Pogradec, and Kukës, were invited for screening, 96.6% of which were screened in the first screening step. (DS)
***Attendance:*** the proportion of the invited infants that have been screened.	*How many infants attended the first screening step and subsequent screening steps after having been invited?*	The percentage of all eligible infants screened in the first screening step was 96.6%. (DS) The percentage of all eligible infants who completed the entire screening protocol was 94.2%. (DS) LTFU from the first towards the second screening step was 33.5%, towards the third screening step 41%, and towards diagnostic assessment 36%. (DS) Note, however, that LTFU differed among screeners, as described under fidelity. (DS) Infants of parents who had to travel longer (not necessarily in rural areas) were more likely to be lost to follow up. (AD)
***Stepwise referral rate:*** the proportion of screened infants that were referred to the next step after a failed outcome. ***Final referral rate to diagnostics:*** the proportion of screened infants that were referred to diagnostic assessment after a failed outcome.	*What proportion of eligible infants was referred for follow-up screening to the second and third screening step after a failed outcome?* *What proportion of eligible infants was referred for diagnostic assessment?*	After the first screening step, 1546 (7%) infants were referred to the second screening step. (DS) After the second screening step, 107 (10.2%) infants were referred to the third screening step. (DS) After the third screening step, 26 (43.5%) infants were referred to diagnostic assessment. (DS) From all three screening steps, 81 (0.35%) infants were referred to audiological diagnostic assessment; 13 after the first screening step, 42 after the second, and 26 after the third. (DS) Referral rates decreased steadily when experience was gained, in maternity hospitals where more infants were born referral rates decreased more rapidly in the first 6 months. (SI, OS, DS [reported in [19]])

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
