# Peer review of "Implementation of Newborn Hearing Screening in Albania"

_2409-515X, 2023, doi:10.3390/ijns9020028_

Round 1

Reviewer 1 Report (Previous Reviewer 2)

Several improvements have been done to the manuscript.

I also wish to thank you the author for their patience in waiting form my late reply (anticipated to the journal due to a very busy time).

I suggest some additional minor revisions.

99 – I would rephrase the sentence “28.000 infants are born in Albania [23], of which 1 to 3 out of every 1000 infants …”; here a suggestion: “About 28.000 infants are born in Albania annually [23], of whom 28 to 84 (1 to 3 out of every 1000 infants) …”

300 – I suggest rephrasing as “1 every 1000 infants” to better show the coherence with the expected incidence reported in line 99. Could you extract the percentage related to NICUs?

301-302 – “These data are described in detail in Figure 3”. Not all the data in the section 292-301 are reported in Figure 1, which only describe the screening process and is not complemented by the diagnostic phase. Please either correct this or extend Figure 3 up to the final diagnosis (and further lost to FU).

Figure 3 (also in coherence with the Table 1) – Several errors; some examples:

1)    Tirana (MG) – 12,412 screened (12,120 + 292), while in the table 12,593 are reported (2337 + + 693 + 2620 + 886 + 1400 + 624 + 2338 + 1695)

2)    Tirana (MG) – 377 (359 + 18) are reported in the figure and 364 (348 + 16) in the Table.

3)    Tirana (KG) – 108 LTFU are reported in the Figure and 109 in the Table

4)    Pogradec – 36 (36 + 0) LTFU are reported in the figure and 38 (37 + 1) in the table

5)    Kukës – 1224 is not equal to 1237 + 13

6)    Total LTFU is 561 (=359 + 18 + 18 + 24 + 36 + 16); I think the difference from the reported 589 is due to the 28 (=19 + 9) whose parents refused. I think this 28 is a “coverage” issue and not a LTFU issue. Nevertheless, if you wish to report these as LTFU, the “refused” boxes should be gray. Please also consider LTFU from screening to diagnosis (29 = 81 – 52).

Table 3 - With respect to my previous suggestion

***

- I suggest to report process indicators as proposed

in https://ijponline.biomedcentral.com/articles/10.1186/s13052-016-0223-1#Sec11  (Additional file 2, section “Monitoring, verifying and reporting phase”) in order to clearly highlight the degree of Universality, Timely detection and Overreferral. This will allow an easier comparison with other programs and benchmarks.

Unfortunately, the intention of this article is not to compare the results of the screening programmes with that of other programmes and benchmarks, but instead report on the implementation of an NHS programme in a country where no such programme exists, not on comparing the fledgling Albanian NHS programme to other, established NHS programmes. Certainly, in the starting phase, the benchmarks as in other longstanding programmes will not be reached and, moreover, LTFU dominated our results with screening to a large extent. Therefore, this is the focus of the article rather than a comparison with existing programmes.

***

Even if I understand your aim, you should not exclude the possibility that further subsequent reviews could include your paper and take your data for comparisons. So, I still believe that a more standardized way of representing process data (e.g., stepwise referral rate) could be useful. For example, I think it is not correct to mix well-born babies and newborns in NICUs as the latter have an order of magnitude higher risk for HL. The suggested reference proposes clearly separating higher and low-risk newborns in process indicator reporting. Timely detection is also a point of interest that could be an interesting element to report.

473 – “Referral rates decreased steadily when screeners gained experience”, please link this conclusion to the result on which in based. In addition, note that this is also reported at 541.

515-16 – I suggest rephrasing as “This means that some infants with hearing loss were likely not diagnosed or did not receive timely intervention” 

532 – “HL” instead of “HI”

539 – I suggest adding that an integrated data repository would also help identifying false-negative subjects, enabling the hospital to identify the need for corrective measures (e.g., re-training of personnel, updating of devices).

Author Response

Please see attached Rebuttal letter 2

Reviewer 2 Report (Previous Reviewer 1)

Please see file attached 

Author Response

Please see attached Rebuttal letter 2

This manuscript is a resubmission of an earlier submission. The following is a list of the peer review reports and author responses from that submission.

Round 1

Reviewer 1 Report

Please  see file attached 

Reviewer 2 Report

I congratulate the authors on their work. I find it very useful for organizations wishing to activate or enhance a universal newborn hearing screening (UNHS) program.

I only suggest a few small revisions to further increase the clarity of the manuscript.

Minor revisions:

-       53 – reference needed;

-       62 – reference needed;

-       Section 2.1 – Preparation and screening protocol. I suggest describing the screening protocols according to the proposed reporting checklist in https://doi.org/10.1186/s12887-015-0404-x (see Table 4) - maybe as supplementary material.

-       Section 2.1 – Preparation and screening protocolYou should mention the guideline you follow to define the screening protocols. You can find a comparison and quality assessment of UNHS guidelines in a recent systematic review: https://doi.org/10.1186/s12887-022-03234-0

-       105 – define the acronyms;

-       124 – define the acronym;

-       138 – please report whether the study (in addition to data collection for health care purposes) was approved by an ethics committee and, if not, the reasons why it was not necessary to involve an ethics committee; 

-       Table 1 – Appropriateness – I think the Operationalisation “Is audiological diagnostic assessment and intervention available to all infants screened positively or referred? “ is a matter of Accessibility;

-       Table 1 – Adoption – I don’t understand the question “How important are infants with a HL for the screeners? “. Do you expect a healthcare worker answering to reply that s/he doesn’t care of a newborn? Please clarify the real meaning of this question.

-       Table 1 – Adoption – I see no link to any issue in the Operationalisation column for the “Database administration went reasonably well given that the programme had just started. (5)” outcome. Maybe “Did screeners integrate NHS in their daily routine”?

-       Table 1 – Fidelity – I would change “Fidelity” with “Compliance”;

-       Table 1 – Fidelity – I think “and did all infants attend follow-up screening in subsequent screening steps? “ is not a matter of compliance (actual actions of operators with respect to the defined procedures) as it may depend from a decision of parents.

-       Table 1 – Coverage – Coverage does not refer to the percentage of newborns “invited for screening (1)” but to the actual percentage of screened newborns. In fact, you correctly refer to coverage in these terms in the lines 267-268.

-       I suggest to report process indicators as proposed in https://ijponline.biomedcentral.com/articles/10.1186/s13052-016-0223-1#Sec11 (Annex 2, section “Monitoring, verifying and reporting phase”) in order to clearly highlight the degree of Universality, Timely detection and Overreferral. This will allow an easier comparison with other programs and benchmarks.

-       223 – Please clarify how the family’s economic status was collected;

-       230 – Please add a reference for the risk factors considered for HL;

-       231-232 – The “total number of infants screened” is a critical number as, from an operational point of view, it can only be considered retrospectively (unless an average number is assumed over a limited past time window). Furthermore, it could include two opposite causal components:

o   The greater the number, the greater the experience of the operators and the quality of the service with a potential consequent reduction in the LTFU;

o   The greater the number, the less time available per infant, with a possible reduction in quality and a potential consequent increase in LTFU.

It could be useful to discuss these aspects. 

-       Figure 2 – I suggest to add a sub-figure to describe the numbers in the section 269 – 276 (i.e., aggregated for all the hospitals). I see a discrepancy between the text and the Figure: I would expect that the sum of the column “Infants recorded in database” to be 22,816 and not 22,656 (=12,813+7780+826+1237).

-       293 – Were dead infants removed from LTFU? 

-       Figure 3 is difficult to understand. The logarithmic scale does not help to compare between screeners (which is the stated purpose of the figure). Since a clearer picture come from Table 2, I suggest to remove Figure 3.

-       461-462 – Define the acronyms.
